# Role of non-specific interactions in the phase-separation and maturation of macromolecules

**Rakesh Krishnan**◉❦, **Srivastav Ranganathan**◉❦*, **Samir K. Maji**◉, **Ranjith Padinhateeri**◉*

Department of Biosciences and Bioengineering, Indian Institute of Technology Bombay, Mumbai, India

❦ These authors contributed equally to this work.
* aditya.sia@gmail.com (SR); ranjithp@iitb.ac.in (RP)

**Data Availability Statement:** The computer code (LAMMPS configuration file, data file) along with the analysis scripts used to produce the data in the paper can be accessed at https://github.com/

## Abstract

Phase separation of biomolecules could be mediated by both specific and non-specific interactions. How the interplay between non-specific and specific interactions along with polymer entropy influences phase separation is an open question. We address this question by simulating self-associating molecules as polymer chains with a short core stretch that forms the specifically interacting functional interface and longer non-core regions that participate in non-specific/promiscuous interactions. Our results show that the interplay of specific (strength, $\epsilon_{sp}$) and non-specific interactions (strength, $\epsilon_{ns}$) could result in phase separation of polymers and its transition to solid-like aggregates (mature state). In the absence of $\epsilon_{ns}$, the polymer chains do not dwell long enough in the vicinity of each other to undergo phase separation and transition into a mature state. On the other hand, in the limit of strong $\epsilon_{ns}$, the assemblies cannot transition into the mature state and form a non-specific assembly, suggesting an optimal range of interactions favoring mature multimers. In the scenario where only a fraction ($N_{frac}$) of the non-core regions participate in attractive interactions, we find that slight modifications to either $\epsilon_{ns}$ or $N_{frac}$ can result in dramatically altered self-assembled states. Using a combination of heterogeneous and homogeneous mix of polymers, we establish how this interplay between interaction energies dictates the propensity of biomolecules to find the correct binding partner at dilute concentrations in crowded environments.

## Author summary

Biological function relies on the ability of biomolecules to bind to specific interaction partners. In the crowded cellular milieu, the process of biomolecules binding to specific interaction partners to carry out a function is non-trivial. A mere diffusion-limited meeting of interaction partners in space does not ensure biomolecular function. Rather, even when in contact, these molecules have to find the correct orientations to carry out their function. Further, for dynamic biomolecules such as polymers, the functional configuration is often one of several possible configurations. In this scenario, assuming the functional configuration after binding involves overcoming a significant entropic barrier. Therefore, the

rakeshkrish/LAMMPS-files The readers can reproduce the results from this paper using the files uploaded in the github link above.

**Funding:** This study was supported by the Department of Biotechnology, Ministry of Science and Technology India, (RP), Grant number: BT/HRD/NBA/39/12/2018-19. The funders had no role in study design, data collection and analysis, decision to publish, or preparation of the manuscript.

**Competing interests:** The authors have declared that no competing interests exist.

interacting biomolecules have to dwell in contact long enough before they reorganize to find the functional orientation! While functional contacts offer an enthalpic gain, they are often only a small fraction of all protein-protein interactions. Therefore, in this paper, we study the role played by non-functional, promiscuous interactions in shaping the thermodynamics and kinetics of the formation of specific interactions. Our results suggest that there exists an optimal range of non-specific interaction strengths, which promotes the process of biomolecular complexes finding the functional configurations.

## Introduction

Liquid-liquid phase separation plays a key role in regulating the Spatio-temporal organization of biomolecules in the cell. Proteins and/or RNA are known to phase-separate into polymer-dense phases known as membrane-less organelles (MLOs), often correlated to external stresses or a certain phase of cell division [1, 2]. The local concentration of proteins within these compartments is significantly higher than their bulk concentrations [3]. One of the potential functions of phase-separated, membrane-less compartments is an increased local concentration of biomolecules facilitating biochemical reactions. In this context, two interaction partners ($\approx$ nM concentrations) meeting in space and locking into place to form functional complexes becomes a non-trivial problem. The low bulk concentration of interacting components makes their diffusing-limited meeting unlikely, but these dynamic molecules must also undergo further structural reorganization to enable key 'native' interactions for their functionality.

Phase-separated MLOs such as P-bodies and stress granules are known to be stabilized by a network of interactions between specifically interacting domains which bind to unique interaction partners. Interestingly, in addition to these specifically interacting domains, protein sequences also bear regions with no known functional interactions. Protein-protein interaction studies suggest that large portions of the interactome are made up of promiscuous or "noisy" interactions with no known function [4–8]. Therefore, could non-specific interactions play a role in the assembly of liquid-like polymer-dense phases en route to the formation of specific contacts? Is there a regime of interaction strengths resulting in non-specifically driven phase separation which maximizes specific contacts at biologically relevant timescales?

Coarse-grained computational models have been extensively employed to study various aspects of biomolecular self-assembly and phase separation. Typical approaches in coarse-grained simulations include the modeling of proteins as lattice polymers [9–11], patchy particles [12, 13] and spheres/beads [14]. Deeds et al. simulated lattice polymers and suggested the existence of a narrow range of temperatures, where the formation of specific complexes is favored despite the presence of promiscuous interactions [9]. Non-specific interactions have also been found to facilitate the formation of ordered, amyloid-like assemblies in patchy-particle simulation studies by Saric et al. [13]. Osmanovic and Rabin employed phenomenological Monte Carlo simulations to demonstrate how non-specific interactions could assist the formation of specific complexes using a group of interacting particles [14]. Another class of computational models is the sticker-spacer model, where the stickers are regions with high attractive interactions and spacer residues, on the other hand, are beads that have excluded volume and weak attractive interactions with other spacers and stickers without any attraction preferences [15, 16]. This model has been effectively employed to study phase separating proteins, and how the valency of stickers (specific interactions) affect the phase separation and aggregation [15]. Most of these existing theoretical models, however, focus on how the interaction parameters influence the thermodynamics of phase separation. In contrast, the dynamics of the self-

assembly process and the role played by non-specific interactions in formation of functional, specific contacts require systematic exploration. [9, 17–20]. In contrast, the process of assembly and the role played by non-specific interactions in the formation of specific polymeric complexes is less clearly understood.

In this work, we study the role of non-specific interactions in driving the phase separation of polymers enabling the specifically interacting domains to form contacts in the polymer-dense phase. Using a coarse-grained representation of proteins, we explore various parameters —specific interaction strengths, non-specific interaction strengths, the ratio of non-specific to specific interaction sites, and systems of different polymer types to understand their role in the phase separation of biological macromolecules. The interacting proteins were modeled as semi-flexible polymer chains consisting of two domains, a small central patch involved in functional interactions (specific interactions) and other regions which participate in weak, promiscuous interactions (non-specific interactions). Off-lattice Langevin dynamics simulations show that non-specific assemblies could be an essential precursor to the formation of functional complexes. We systematically explore the interaction parameter space to identify the regime that maximizes contact between specifically interacting sites. We further tuned the ratio of specific and non-specific interaction sites and probed the critical interaction strengths resulting in optimal specific contacts. Using an extension of this model, we demonstrate how the transition between non-specific to specific interaction-dominant assemblies is influenced by the presence of non-interacting polymer chains (crowders) and (exclusively) non-specifically interacting polymers in the system. Overall, our results shed light on how interactions involving noisy regions could be tuned to promote the formation of otherwise key, low-probability functional interactions [11, 21, 22].

## Model

### Modeling intrinsically disordered proteins as semi-flexible polymer chains

We model proteins as bead spring polymer chains, each chain of 80 beads in length. Each polymer bead is an effective interaction unit in the model that maps onto 2–3 amino acids in an intrinsically disordered protein [23]. The 80 bead polymer chain in our simulations mimics a typically sized multivalent protein with a high degree of disorder (and conformational flexibility). This length scale is of the same order as the mean length of intrinsically disordered proteins in the disprot server [24]. The self-assembling coarse-grained proteins are composed of two types of beads (Fig 1)—blue beads that model weak non-specifically interacting residues and a cluster of red beads, which mimic a specific interaction patch within the protein. We further employed a three-bead polymer model to systematically vary the fraction of specific to non-specifically interacting residues in a polymer chain (Fig 1B, three-bead model).

We consider interactions involving red beads in the polymer exclusive (and specific) because they interact with the other red beads with higher attractive interactions (specific interactions) than red-blue or blue-blue interactions, which are weakly attractive in comparison (non-specific interaction). The strength of non-specific interactions (blue-blue and blue-red beads) is defined as $\epsilon_{ns}$, while the specific interaction strength is $\epsilon_{sp}$. In our model, we placed specifically interacting red beads in the center of the polymers. The choice of a centrally located specific interaction region was to model functional patches buried within a protein and to maximize the net specific interaction strength by clustering the residues in close contact [25]. We used Langevin dynamics or Replica Exchange Langevin Dynamics (RELD) methods to simulate the system of polymers (see Methods section for details of the simulation methods).

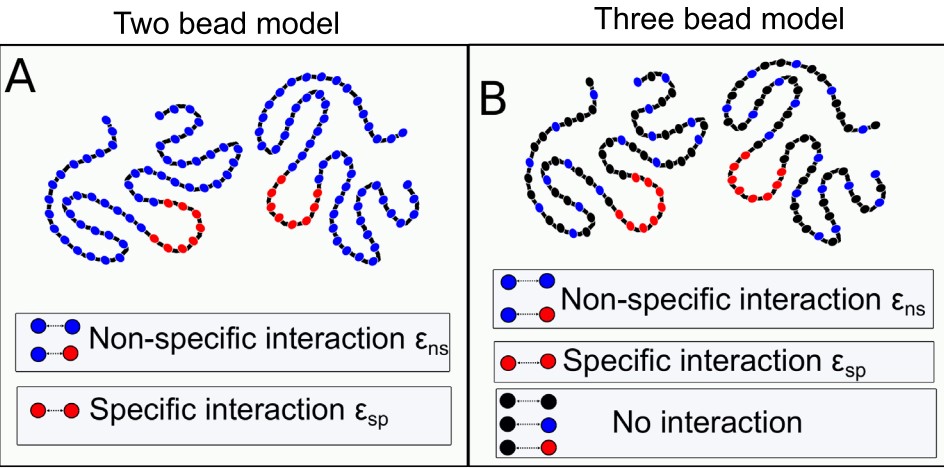

**Fig 1. Bead-spring polymer model of protein.** A) Chains consist of a central "functional" patch (red) surrounded by other beads which make up non-specifically interacting regions (blue). B) Three bead polymers with black beads representing non-interacting residues (only repulsive interaction), in addition to specifically (red) and non-specifically (blue) interacting residues. Specific and non-specific interaction strengths are represented by $\epsilon_{sp}$ and $\epsilon_{ns}$, respectively, with $\epsilon_{sp} > \epsilon_{ns}$. We employ the 3-bead model to vary the fraction of non-specifically interacting residues, while keeping the polymer length fixed.

## Biological motivation behind the model

The prime focus of this study is to probe how promiscuous interactions could influence the formation of specific contacts between polymer chains. Therefore, we primarily employ the model wherein the specifically interacting sites are clustered in close proximity within the polymer chain. In a recent study by Holehouse et al, closely clustered specific interaction sites have been shown to reduce saturation concentration for phase separation [25]. In our simulations, each specific interaction residue (red beads) can interact with multiple beads as allowed by steric hindrance. Therefore, the effective valency per specifically interacting bead (red beads) would be of the order of the coordination number for the beads (valency > 1). However, despite the relatively stronger specific interactions involving red beads, these residues make up a very small fraction of the polymer chain. This is in tune with interactome studies that suggest that >50% of protein-protein interactions do not code for any known functional contacts in protein complexes. Therefore, the non-specific interactions in these polymers are entropically favored, while specific interactions offer an enthalpic gain. In this study, we attempt to identify how non-specific interactions can tune the propensity of polymer chains to form these entropically disfavored specific interactions.

## Range of interaction strengths

A well-studied specific interaction across several signaling proteins and in the context of phase-separating systems is the SH3-PRM interaction. In their monomeric form, the dissociation constants for the SH3-PRM interaction is 350 uM, which is equivalent to an interaction strength of the order of a few (5–10 $k_BT$) $k_BTs$ [26]. In the seminal work by Harmon et al., the critical interaction strengths for SH3-PRM interactions in the spacer-sticker model were reported to be 3 $k_BT$ [16]. The 'specific interactions' involving red beads in our coarse-grained polymer system, therefore, mimic such functional inter-domain interactions in proteins and are varied within the ranges of interaction strengths discussed above. On the other hand, the non-specific interactions are weak, short-range interactions between different amino-acid

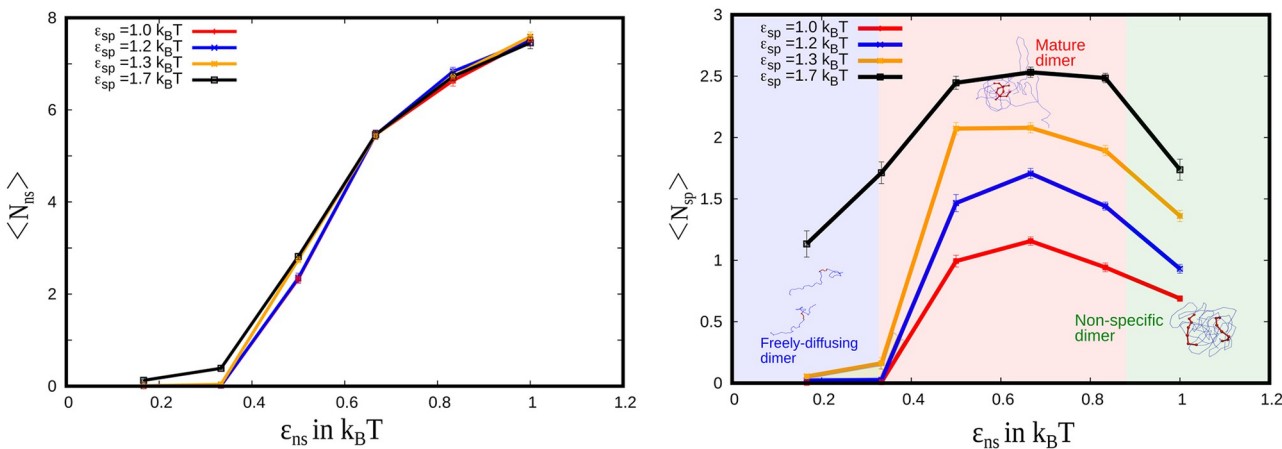

**Fig 2. State of dimers for different non-specific interaction strengths.** A) Average non-specific contacts per bead as a function of $\epsilon_{ns}$ for different value of $\epsilon_{sp}$. B) Average specific contacts per bead as a function of $\epsilon_{ns}$ for different value of $\epsilon_{sp}$. Both are averaged over 100 trajectories (colors are just a guide to eye).

pairs in intrinsically disordered proteins in an aqueous solvent. Dignon et al. studied the phase separation of several known IDR-containing proteins such as LAF and FUS and reported these interaction strengths to be of the order of 0.2 to $1k_BT$ (amino-acid pairwise) [27]. We, therefore, vary the strength of non-specific interactions in our study within this aforementioned range.

## Order parameters used in the paper

In order to track the process of functional dimer, multimer formation we primarily employ the following order parameters. To define the state of the polymeric system, we employ the order parameter $N_{ns}$ and $N_{sp}$, inter-chain non-specific and specific contacts respectively (Figs 2, 3 and 4). Any two beads from different polymer chains that are within an interaction cutoff of 2.5 $\sigma$ are considered to be in contact. Any contact involving blue beads is referred to as a non-specific contact while those between two red beads from different chains is labeled a specific contact. To distinguish between non-specifically and specifically interacting dimers/multimers, we employ the radius of gyration of specifically interacting residues (red beads), $R_g^{norm}$—as the order parameter (Fig 4). $R_g^{norm}$ is normalized by the number of polymers in the system ($N_{lc}$). We also compute local density profile of specific interaction sites to characterize how specific interaction sites are distributed spatially within the structure.

To study the spatial distribution of red beads (Fig 5), we draw many spherical shells starting from the center of mass of the system (center of the multimer). We compute the fraction of red beads in shells defined by spheres of inner radius $R_1$ and outer radius $R_2$, where $R_2-R_1 = 3$Å. The fraction of red beads within the shell is defined as the number of red beads in the shell divided by the total number of red beads in the cluster when a multimer is formed. Otherwise, for a freely diffusing state, it is normalized with the total number of red beads in the system (see Fig 5A). In Fig 5B, we have also computed the number density of red beads defined as $\phi_{red} = \frac{N_{red}}{(4/3)\pi(R_2^3 - R_1^3)}$, where $N_{red}$ is the number of red beads within the probe shell. The order parameter notations and the physical quantity that they refer to are listed in Table 1.

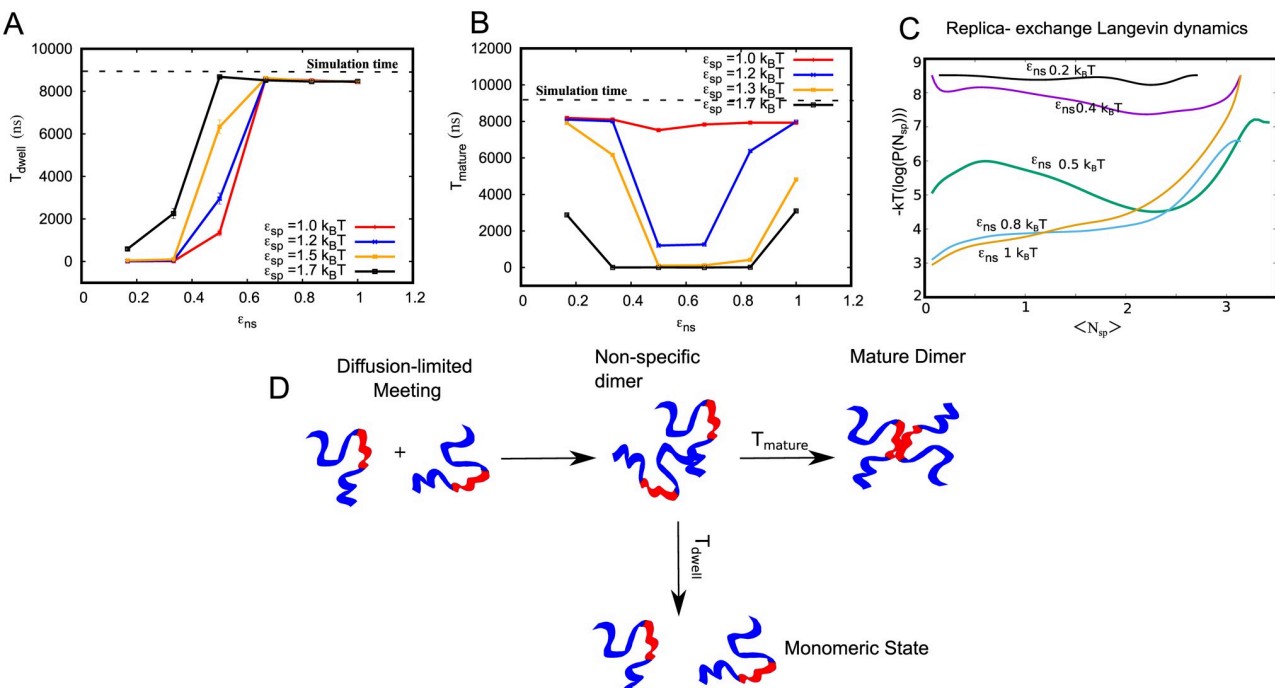

**Fig 3. Key timescales characterizing maturation dynamics.** A) Mean dwelling time in the dimeric state ($T_{dwell}$) as a function of non-specific interaction strength ($\epsilon_{ns}$) and different $\epsilon_{sp}$. B) Mean maturation time, reorganising time from non-specific dimer to mature dimer as a function of $\epsilon_{ns}$ for different values of $\epsilon_{sp}$ (smoothed solid curves are a guide to eye). C) Free energy as a function of specific contacts per bead computed using replica exchange langevin dynamics simulations. Individual free energy curves represent simulations performed at different values of $\epsilon_{ns}$. For an $\epsilon_{ns} = 0.5 k_BT$ (green curve), there is a sharp minima for higher value of specific contacts per bead. For high and low values of $\epsilon_{ns}$ this minima is not observed, suggesting that there is an optimal value of $\epsilon_{ns}$ at which functional dimers are favored. D) Schematic representation of the key events en route to forming a mature dimer. Interaction partners meet at timescales that are diffusion-limited. Upon initial meeting, the dimers are in contact due to non-specific interactions. This non-specific dimer could then either diffuse away at a typical timescale of $T_{dwell}$ or reorganize to form a mature, functional dimer with a characteristic reorganisation time of $T_{mature}$.

### Kinetic order parameters

$T_{dwell}$, the dwelling time, is the time spent by the polymers in the dimeric state before dissociating to the monomeric state. To compute $T_{dwell}$, we simulated 100 independent trajectories for the dimeric system and kept track of the time spent by polymers in the dimeric state before dissociating away. In the paper, we report the mean time spent in this dimeric form (with the simulation timescale being the upper limit) averaged across several such dissociation events across 100 trajectories (error bars shown in Fig 3A). If any part of the two polymer chains is within a 2.5 $\sigma$ cutoff, they are considered in contact with each other. $T_{mature}$ is the maturation time—the time taken for non-specific dimers to reorganize into the mature dimeric state. To compute $T_{mature}$, we compute the mean time taken to go from the non-specifically stabilized form to a functional dimer. For the purpose of these calculations, we define a dimer as a 'non-specific dimer', the first instance that they come in contact with each other and engage in less than ten specific contacts with each other. The time taken to go from this non-specific dimeric state ($N_{sp} < 10$) to a mature dimeric state ($N_{sp} >= 10$) is then recorded. It must, however, be noted that we consider the mature dimeric configuration to be stable if the structures remain in this configuration for at least 20 ns. The mean $T_{mature}$ reported in this study were averaged over 100 independent trajectories for each combination of interaction parameters.

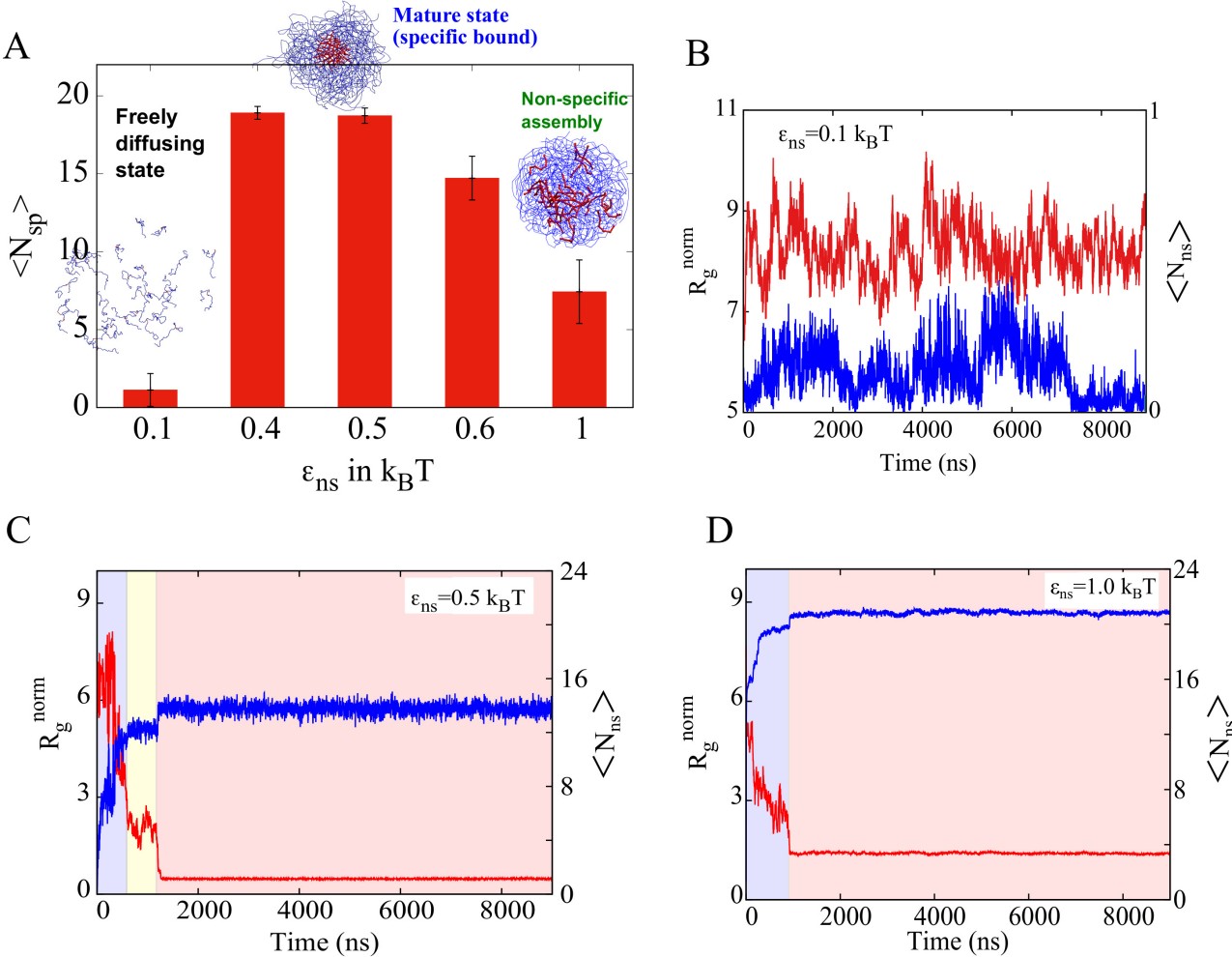

**Fig 4. State of multimeric system for varying strengths of $\epsilon_{ns}$.** A) Average number of specific interaction per bead (red-red) $N_{sp}$ for different of non-specific interaction strength $\epsilon_{ns}$ values. (B,C,D) Radius of gyration ($R_g^{norm}$) and Number of non-specific interactions per bead ($N_{ns}$) are plotted as a function of time for $\epsilon_{sp} = 1.2$ k$_B$T. B) For weak $\epsilon_{ns} = 0.1$ k$_B$T we observe neglibile non-specific contacts in the system ($N_{ns} \approx 0$). This is accompanied by large values of $R_g^{norm}$ suggesting that the stable multimers are not observed. C) For $\epsilon_{ns} = 0.5$ k$_B$T, there is an initial regime of low $N_{ns}$ and large $R_g^{norm}$ while the polymer chains are yet to form multimeric clusters. This is followed by a regime (shaded yellow) where there is an increase in $N_{ns}$ suggesting the formation of multimers stabilized by non-specific interactions. This increase in $N_{ns}$ further enables the central functional patches to come in contact, indicated by the further decrease in $R_g^{norm}$ (red regime). D) A further increase in $\epsilon_{ns}$ to 1 k$_B$T results in the initial formation of non-specifically stabilized multimers without a further reorganization into a functional complex where the central red-patches do not phase-separate into a separate region. At these high values of $\epsilon_{ns}$, the multimers forms a non-specific assembly.

## Results

In this paper, we systematically explore the interplay between specific and non-specific interactions in the formation of functional polymeric complexes. In the first part of the paper (Figs 2 and 3), we explore how the energetics of interaction can shape the process of dimer formation (Figs 4–8). We further extend this model to study how non-specific interactions shape oligomerization. In the final part of the paper, we study the multimerization process for systems heterogeneous mix of polymers.

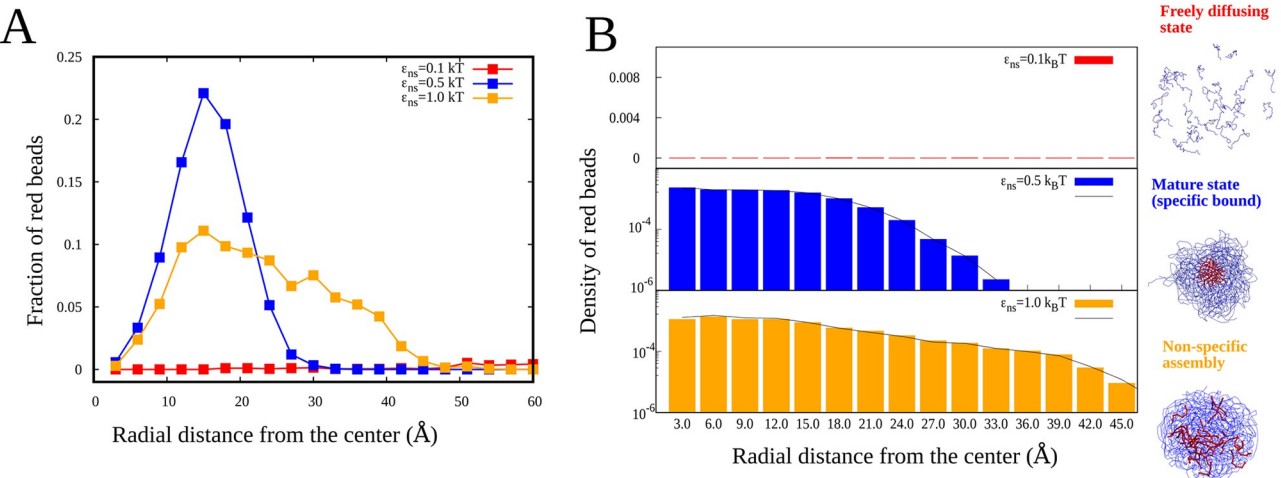

**Fig 5. Spatial distribution of specific interaction sites.** A) Fraction of red beads is each shell is plotted as a function of radial distance from the center of the cluster (blue and orange curves for self-assembled states) or the simulation box (freely diffusing state, redcurve). B) Density of red beads in each shell is plotted as a function of radial distance from the center.

## Proteins as dynamic polymers: Role of non-specific interaction in the formation of functional dimers

Cellular function hinges on biomolecular recognition and the formation of specific protein-protein complexes. Interestingly, interface residues involved in stabilizing biomolecular complexes account for only a tiny fraction of protein surfaces [4, 5, 7]. Here, we first probe how the interplay between specific and non-specific interactions can shape the process of dimerization with coarse-grained polymer chains (Fig 1A). In order to characterize the configuration of the dimer, we use specific contacts per bead (contacts involving red beads only, $N_{sp}$) and non-specific contacts per bead (contacts involving blue beads, $N_{ns}$) as the order parameter. Dimers, where the central red patches are in contact, are referred to as mature dimers. This mature dimeric state is similar to protein-protein complexes stabilized in the correct configurations

**Table 1. Parameters abbreviations in simulations.**

| Parameters | Physical Quantity |
|---|---|
| $\epsilon_{ns}$ | Non-specific interaction energy (pairwise) |
| $\epsilon_{sp}$ | Specific interaction energy (pairwise) |
| $<N_{ns}>$ | Average number of non-specific contacts per bead |
| $<N_{sp}>$ | Average number of specific contacts per bead |
| $N_{frac}$ | Fraction of non-specifically interacting residues (given in percentage) |
| $R_g^{red}$ | Radius of gyration of red beads |
| $N_{lc}$ | Number of polymers in largest cluster. |
| $R_g^{norm}$ | Radius of gyration normalised |
| $T_{dwell}$ | Dwelling time |
| $T_{mature}$ | Maturation Time |
| $\phi_{red}$ | Number density of red beads |

that enable biomolecular function. On the other hand, dimers that are stabilized by non-specific contacts alone are referred to as non-specific dimers.

To study the interplay between specific and non-specific interactions, we vary $\epsilon_{ns}$ and $\epsilon_{sp}$ systematically and investigate the process of formation of mature dimers. For each value of $\epsilon_{ns}$ and $\epsilon_{sp}$ shown in Fig 2, we simulated 100 independent trajectories for the dimeric system for a timescale of 9 $\mu$s (per trajectory). In Fig 2A and 2B, we plot the average value of non-specific contact per bead ($\langle N_{ns} \rangle$) and specific contact per bead ($\langle N_{sp} \rangle$) respectively as a function of varying $\epsilon_{ns}$, for different values of $\epsilon_{sp}$. At low values of $\epsilon_{ns}$, where both $\langle N_{ns} \rangle$ and $\langle N_{sp} \rangle$ are low, the system remains in a monomeric state for the range of values (1 $k_B$T to 1.7 $k_B$T) of $\epsilon_{sp}$ that we have studied (shaded leftmost region in Fig 2A and 2B). As we increase the strength of non-specific interaction ($\epsilon_{ns} > 0.4$ $k_B$T), we observe an increase in $\langle N_{ns} \rangle$ (shaded middle region in Fig 2A). Interestingly, this regime of $\epsilon_{ns} > 0.4$ $k_B$T, where there is an increase in $\langle N_{ns} \rangle$, is also accompanied by an increase in $\langle N_{sp} \rangle$, (shaded middle region in Fig 2B) suggesting that the dimers undergo maturation. A further increase in $\langle N_{ns} \rangle$ at $\epsilon_{ns} > 0.8$ $k_B$T, however, results in a non-specific dimeric state with a corresponding decrease in $\langle N_{sp} \rangle$ (shaded rightmost regions in Fig 2A and 2B). In this regime of strong non-specific interactions, the system fails to transition into the mature dimeric state and remains in a non-specifically bound state.

In order to understand the mechanism by which non-specific interactions modulate this transition to the mature state, we investigate the key timescales of dimer assembly in detail. We first computed the average time that a dimer remains stable before diffusing away to a monomeric state ($T_{dwell}$). The dwelling times were calculated across multiple binding-unbinding events over 100 independent trajectories. It must, however, be noted that the upper bound for the timescales in our calculation is the length of the trajectory itself (9$\mu$s). In Fig 3A, we plot the mean dwelling times as a function of non-specific interaction strength. At low values of $\epsilon_{ns}$, the dwelling times are very low, suggesting that the two polymer chains remain in a dimeric state for very short times. As we increase $\epsilon_{ns}$, there is a significant increase in $T_{dwell}$. At large values of $\epsilon_{ns}$, the dimers remain in contact for $T_{dwell}$ values that are comparable to the simulation timescale. While the increase in $T_{dwell}$ as a function of $\epsilon_{ns}$ shows the increased stability of the dimeric state, it does not provide insights into the ability of the dimers to undergo a transition into the mature state. In order to probe the maturation process, we compute the average time taken for the dimers to transition from the non-specifically bound state to the mature state ($T_{mature}$). For weak non-specific interactions, the system never undergoes maturation, and hence $T_{mature}$ values approach the timescale of the simulation (horizontal dashed line) in Fig 3B. As we increase $\epsilon_{ns}$, we observe a decrease in maturation times, suggesting that the stabilization of the dimeric state due to non-specific contacts allows the polymer chains to dwell long enough in the vicinity before undergoing reorganization to the mature state. However, as we increase $\epsilon_{ns}$ above 0.6 $k_B$T, we see an increase in maturation times. At these high values of non-specific interaction, there is a slowdown in maturation since the dimers get trapped in the non-specifically bound state. These results suggest that there exists an optimal range of $\epsilon_{ns}$ wherein maturation of non-specific dimers to specifically bound dimers occurs at fast, biologically relevant timescales.

To further understand how $\epsilon_{ns}$ could modulate both the equilibrium state as well as the kinetics of this transition, we performed parallel tempering (replica exchange) Langevin dynamics simulations to construct the free energy profiles for the dimeric system. We simulated 24 different replicas with fixed $\epsilon_{sp} = 1.2$ $k_B$T, each spaced 4 K apart for a temperature range of 309–401 K. The free energy curves in Fig 3C show that for very low values of $\epsilon_{ns}$, we do not observe a clear minimum at higher values of $N_{sp}$ (specific contacts per bead). Also, for very large values of $\epsilon_{ns}$, we observe a steep barrier ($>3$ $k_B$T) to go from low to high values of specific contacts, suggesting that the system would prefer to remain in a non-specifically

bound state. However, at an intermediate value of $\epsilon_{ns}$, we see a defined minimum for the higher value of $N_{sp}$. This indicates that non-specific dimers can transition to mature, specifically bound dimers in this regime. The enthalpic gain from non-specific interactions in this regime, therefore, offsets the entropic cost of assuming the mature form that maximizes specific contacts.

Overall, these results suggest that the meeting of functional interfaces (red patch) between two polymer molecules must be facilitated by weak $\epsilon_{ns}$ that increases the dwelling time of a non-specifically bound dimer. The individual chains within the dimer can then undergo rearrangement, eventually resulting in mature dimers with the central specifically interacting patches in contact with each other (see Fig 3D).

## Functional biomolecular complexes: Non-specific interaction, phase separation and maturation

Phase-separated biological assemblies such as membrane-less organelles can often localize hundreds of molecules within them [28–32]. The concentrations of proteins within these clusters are several-fold higher than their bulk concentrations, facilitating favorable biomolecular interactions within the dense phase. Also, promiscuous interactions have been attributed to stabilizing the phase-separated state while allowing specific interactions to form within the polymer-rich phase. In this context, we simulated 30 polymer chains, each having two patches with the same behavior as described so far (see Fig 1A), in a box such that the concentration is $\approx 200\mu$M. We systematically probe the role of the non-functional regions of the polymer chain in driving phase separation that enables the meeting of functional patches within the polymer-dense phase.

For $\epsilon_{sp}$ of 1.2 $k_BT$, there is negligible development of contacts between specific patches, at low values of $\epsilon_{ns}$ of 0.1 $k_BT$ (Fig 4A). As we increase the $\epsilon_{ns}$ = 0.1 $k_BT$ to 0.4/0.5/0.6 $k_BT$, we see a significant increase in the $N_{sp}$ resulting in a mature complex, where all red patches come together. However, as the strength of the non-specific interactions approaches that of the specific interactions ($\epsilon_{sp}$ = 1.2 $k_BT$ and $\epsilon_{ns}$ = 1 $k_BT$), there is a sudden decrease in the number of specific contacts suggesting a slow down in the search process of the specific interfaces. As $\epsilon_{ns}$ approaches 1 $k_BT$, the polymer chains form a non-specific assembly with non-specific interactions outnumbering and thus outcompeting specific contacts. Overall, consistent with the dimer simulations, despite the relatively higher strength of the interaction between the red patches, the ability to phase separate and form a mature multimer is modulated by the 'non-specific' regions of the polymer chain. Representative snapshots indicating the nature of the self-assembled state can be seen in Fig 4A.

To further understand the kinetics of red patches finding each other with the help of non-specific blue beads, we tracked two parameters as a function of time: $N_{ns}$ (Number of non-specific contacts per bead) and $R_g^{norm}$ (Radius of gyration of red beads, $R_g^{norm} = R_g^{red}/N_{lc}$) as in Fig 4B, 4C and 4D. For very weak non-specific interactions (0.1 $k_BT$, Fig 4B) there is no development of non-specific contacts (blue curve) as represented in the $y_2$ axis (the Y-axis on the right side). At the same time, the $R_g^{norm}$ is very large (red curve, marked on the $y_1$ axis—the major Y-axis on the left side), suggesting that the red beads are far away and are not in contact. In contrast, in the optimal $\epsilon_{ns}$ regime (0.5 $k_BT$) that favors the development of specific contacts (see Fig 4A), the early part of the simulation is characterized by a rapid increase in non-specific contacts (Fig 4C, blue curve, shaded blue region). This initial increase in $N_{ns}$ results in non-specifically bound phase-separated structures. Interestingly, following the initial increase and saturation of $N_{ns}$, the system dwells in this non-specifically bound state (Fig 4C, yellow regime). As evident from the radius of gyration of the red beads, upon saturation of non-

specific interactions (Fig 4C, blue regime), the clusters undergo reorganization to maximize specific contacts, as seen from a further collapse in the $R_g$ curve in Fig 4C (yellow regime). While our simulations were performed with specific interaction sites clustered at the center of the polymer, these results hold true for other configurations. (S1 and S2 Figs).

Our simulations reveal that, in different regimes of $\epsilon_{ns}$, the multimeric system could end up in distinct structural states. To quantify differences in the spatial distribution of specific interaction regions within the self-assembled multimers, we plot the fraction of red beads and the number density of red beads (see Order Parameters subheading under the Model section) in each probe shell of increasing radii from the center of mass of the system. In Fig 5A, we show the fraction of red beads in each shell plotted over the radial distance from the center. For the $\epsilon_{ns} = 0.1$ $k_BT$, the fraction red beads is very low– a flat curve is observed (red curve, Fig 5A). Similarly, the density of red beads is also very low, indicating that the red beads are spread across the simulation box (top panel in Fig 5B). When the $\epsilon_{ns}$ is 0.5 $k_BT$, the fraction of red beads is highly peaked compared to $\epsilon_{ns} = 1.0$ $k_BT$ (orange curve), which is more flat. When the density is plotted for $\epsilon_{ns} = 0.5$ $k_BT$, we find a constant density for small distances and decays quickly, suggesting that the red beads are concentrated at the center (see Fig 5B, middle panel). For $\epsilon_{ns} = 1.0$ $k_BT$, the density is nearly uniform throughout the cluster, suggesting that red beads are found everywhere (see Fig 5B, bottom panel).

To further establish the narrow range of $\epsilon_{ns}$ in which the mature complex is favored, we also performed RELD simulations for the system with 30 polymeric chains with 24 replicas, with temperatures ranging from 309 K to 401 K. We plot the resulting free energy profiles for the 30-mer system at a statistical temperature of 309 K, with $N_{sp}$ as the order parameter used to identify the transition to the mature state with solid-like core (Fig 6). Consistent with our RELD free energy profiles for the dimeric system, for weak $\epsilon_{ns}$ in the multimeric system, we do not observe defined minima at high values of $N_{sp}$. Also, for extremely strong $\epsilon_{ns}$ (Fig 6D and 6E), we observe a sharp free energy minimum for low values of specific contacts, suggesting that the system remains in a non-specifically assembled state and does not access the mature state in such a scenario. Interestingly, for an intermediate value of $\epsilon_{ns} = 0.4$ and 0.6 $k_BT$, we observe a defined minima for large specific contact numbers establishing that the mature state gets facilitated in such a regime of $\epsilon_{ns}$ (see Fig 6B and 6C). Overall, these results suggest that phase-separation followed by maximal specific contact formation is favored in a narrow regime of $\epsilon_{ns}$, and any perturbation to either side would result in dramatically different equilibrium and dynamic properties.

## Selective recruitment of polymer chains from a heterogeneous mixture

Our results so far establish the role of promiscuous interactions in driving the formation of mature dimers and multimers. However, these simulations were performed for a homogeneous mixture of polymers where all polymer chains can interact with each other, specifically and non-specifically. However, the cellular milieu is crowded and is constituted of a heterogeneous mix of biomolecules that can interact non-specifically to varying degrees. Could polymer chains that are unable to participate in promiscuous interactions get recruited into the multimeric complexes? To address this question, we first perform simulations wherein we introduce polymer chains that can engage in specific interactions alone but lack the ability to interact non-specifically (System 1 in Table 2 and Fig 7A). We refer to these non-specifically inert but specific-interaction compatible polymer chains as Type 2 polymers. The non-specific (blue beads), and specific interaction (red beads)-compatible polymer chain employed in simulations so far, is referred to as a Type 1 polymer. Interestingly, we observe that there is negligible recruitment of Type 3 chains in multimeric clusters, which are enriched in Type 1 chains

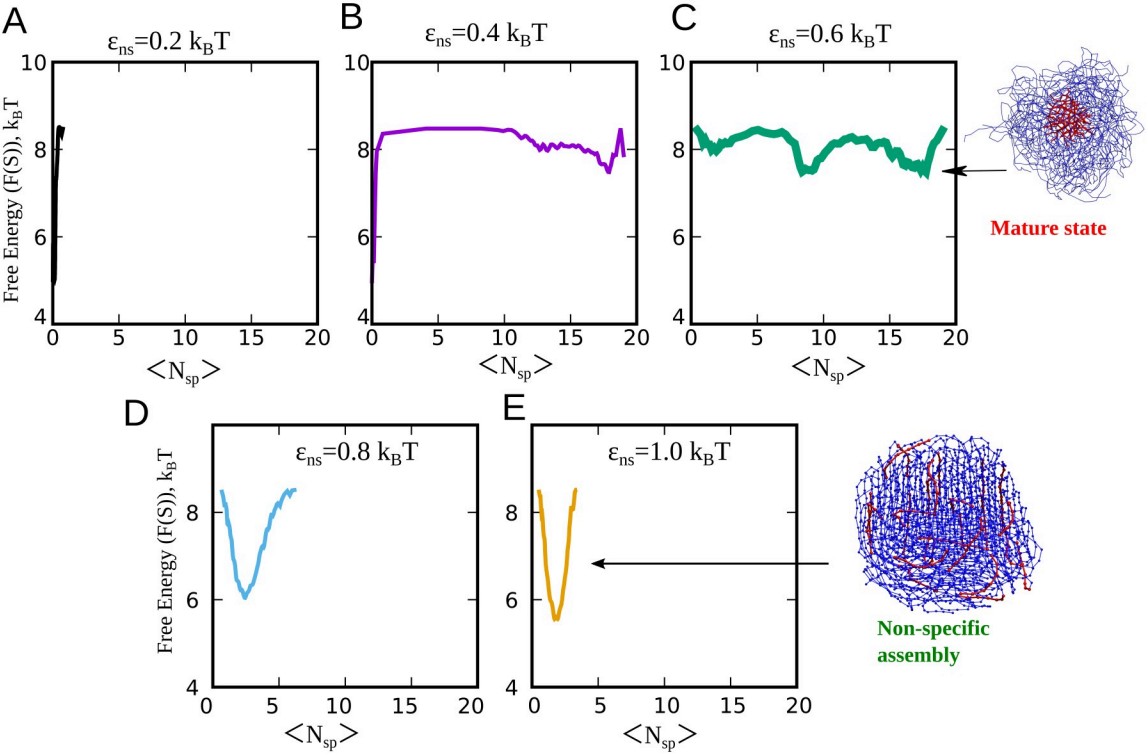

**Fig 6. RELD simulations for the multimeric system.** Free Energy profiles as a function of specific contacts as the order parameter. The free energy curves in A, B, C, D, E correspond to $\epsilon_{ns}$ values of 0.2, 0.4, 0.6, 0.8 and 1 $k_B$T, respectively. For $\epsilon_{ns}$ = 0.2 $k_B$T, the system remains predominantly in a monomeric state. For an intermediate value of $\epsilon_{ns}$ = 0.4 and 0.6 $k_B$T, we observe a sharp minima at higher values of specific contacts (per red bead), indicating that the system undergoes a transition to the mature state at this value of non-specific interaction strength. For $\epsilon_{ns}$ higher than 0.6 $k_B$T, the system forms a non-specifically assembly, as evident by the sharp minima for low values of specific contacts.

(see snapshots in Fig 7B and 7C). This selective recruitment of Type 1 polymers, despite the Type 2 chains harboring a specific interacting cluster of residues, highlights the critical role played by promiscuous interactions in facilitating the formation of specific contacts. In the second scenario (System 2 in Table 2 and Fig 7A), we introduce Type 3 polymer chains that can engage in specific interactions with Type 1 polymers but are unable to engage in non-specific

**Table 2. Heterogeneous mix of polymer definition and interaction involved in them and number of polymers in the system.**

| System names | Different types of polymers in simulation box | Number of polymers |
|---|---|---|
| **Default** | **type 1** polymer with blue and red beads having the following interactions. Blue-blue: $\epsilon_{ns}$ = 0.5$k_B$T, blue-red: $\epsilon_{ns}$ = 0.5$k_B$T, red-red: $\epsilon_{sp}$ = 1.2$k_B$T | 30 chains of Type 1 |
| **System 1** | **type 1** as above; **type 2** polymer with black and red beads having following interactions black-black: repulsive LJ, black red: repulsive LJ, red-red: $\epsilon_{sp}$ = 1.2$k_B$T, blue-black: repulsive LJ | 15 chains of Type 1 + 15 chains of Type 2 |
| **System 2** | **type 1** as above; **type 3** polymer with green beads and red beads with following interactions green-green: $\epsilon_{ns}$ = 0.5$k_B$T, green-red: $\epsilon_{ns}$ = 0.5$k_B$T, red-red: $\epsilon_{sp}$ = 1.2$k_B$T and green-blue: repulsive LJ. | 15 chains of Type 1 + 15 chains of Type 3 |

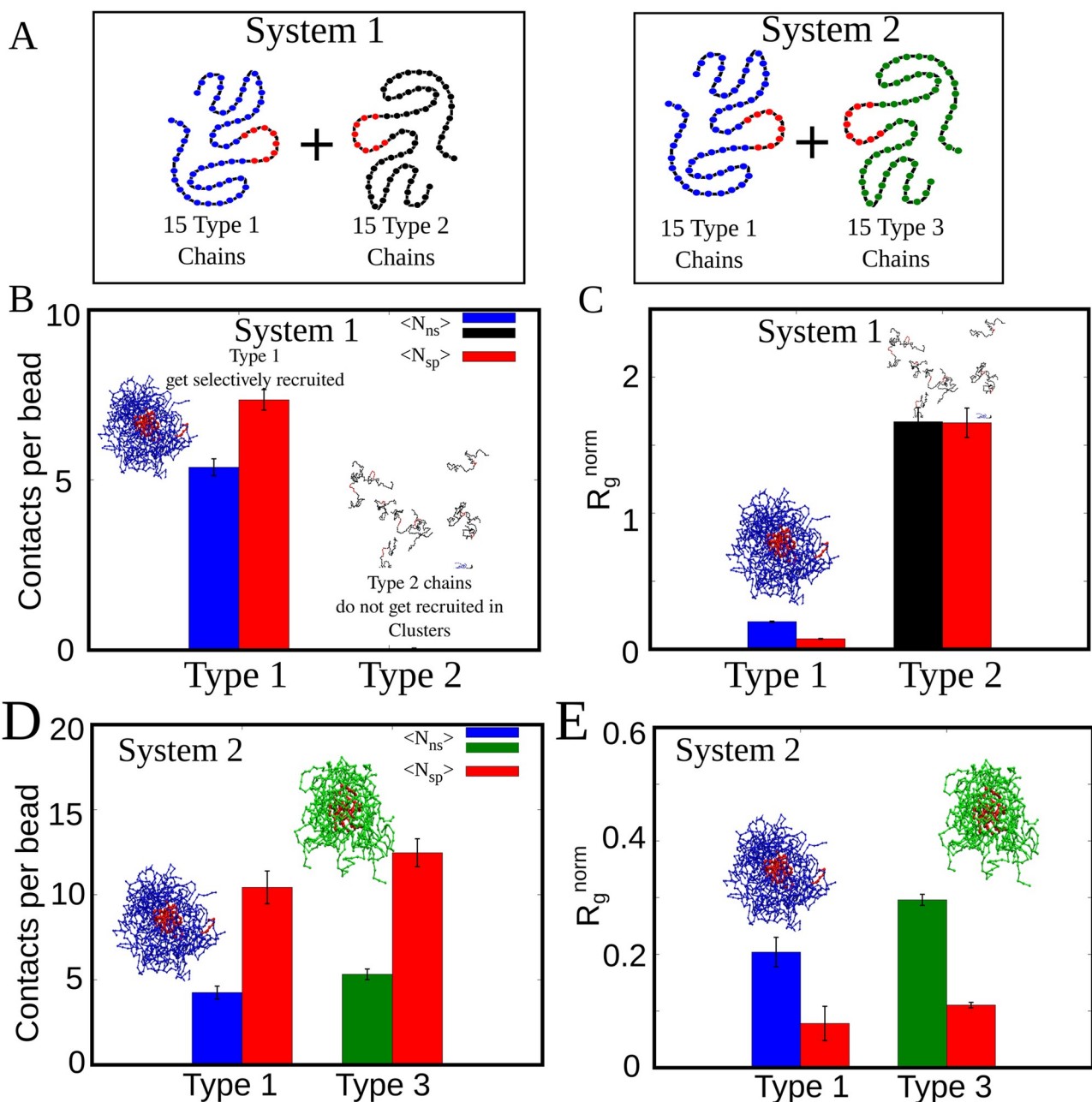

**Fig 7. Selective recruitment of polymers in 2-component mixtures.** A) Schematic representation of the two simulation scenarios and the polymer types in both cases. B) Non-specific and specific contacts per bead for Type 1 and Type 2 polymers for simulations involving System 1. Type 2 polymers unable to participate in non-specific interactions show no recruitment in clusters, and hence do not develop non-specific contacts. C) Radius of gyration of red beads in Type 1 and Type 2 polymers. Red beads in type 1 polymers get condensed within the cluster while Type 2 polymers show large $R_g$. D) Non-specific and specific contacts per bead for Type 1 and Type 3 polymers showing two distinct clusters for this scenario. E) Radius of gyration of red beads in Type 1 and Type 3 polymer chains for System 2 simulations.

interactions with Type 1 chains. However, these Type 3 chains can engage in non-specific contacts with other Type 3 chains. In this scenario, we observe two distinct multimeric clusters, each enriched in Type 1 and Type 3 polymer chains selectively (Fig 7D and 7E). These results, therefore, highlight that promiscuous interaction could have been tuned evolutionarily to

achieve selective enrichment of biomolecules within phase-separated clusters while conserving specific interaction domains.

## Three bead polymer model: Non-specific regions drive mature multimers in a very narrow parameter regime

The simulations so far dealt with the self-organization of a homogeneous system of polymer chains made up of two kinds of beads. The results demonstrate how the interactions between the non-specific region can act as a handle for tuning the timescale of multimer formation and phase separation-like behavior of biopolymers. In our simulations so far, the whole length of the polymer chain can involve in weakly attractive non-specific interactions. However, the molecular language of proteins composed of 20 different building blocks (amino acids) with different interaction propensities enables a more subtle tuning of the self-assembly/phase-separation phenomenon. We, therefore, employ a three-bead model (Fig 1) wherein we systematically vary the fraction of residues that can engage in attractive, short-range, non-specific interactions. In this three-bead model, we introduce an inert monomer type (black beads in Fig 1B), which occupies volume but does not participate in attractive interactions. For a fixed value of specific interaction strength of $\epsilon_{sp} = 1.2\ k_BT$, we systematically vary the strength of non-specific interaction $\epsilon_{ns}$ and fraction of non-specifically interacting residues ($N_{frac}$). This systematic exploration allows us to probe whether non-specific interactions can promote specific contacts even when there are fewer interaction sites scattered across the polymer.

At a very low fraction of non-specifically interacting residues $N_{frac}$, the polymers do not engage in non-specific contacts. In this parameter regime of $\epsilon_{ns}$ and $N_{frac}$ which does not promote non-specific interactions (Fig 8E, black dots), we observe negligible development of specific contacts (Fig 8F, black dots). Further, in the parameter regime demarcated by the dotted boxed region in Fig 8E, wherein non-specific contacts start appearing, the polymer assemblies also form functional contacts (Fig 8F, grey shaded region), suggesting that the mature multimeric state is favored in this regime. (see simulation snapshot in Fig 8C). However, for extreme values of $\epsilon_{ns}$ and $N_{frac}$, there is a reduction in functional contacts (Fig 8F, yellow shaded region) and the system remains in a non-specifically bound state (see snapshot in Fig 8D). In this regime, where non-specific interactions outcompete specific interactions, there is no significant enthalpic gain (compared to the entropic loss) in forming mature multimers. These results corroborate the two-bead model's findings wherein a critical non-specific interaction strength is vital for self-assembly en route to the development of specific contacts.

## Discussion and conclusion

### Biological motivation for the simulations

Biological reactions rely on the ability of biomolecules to not only bind to the correct binding partners but also assume the correct 'functional' configurations. However, the concentration of biomolecules like proteins is typically very low ($\approx$ nM) within cells, making the diffusion-limited reactions malleable to tuning by external factors. The cellular environment is crowded with different types of biomolecules diffusing and promiscuously interacting with each other [9]. Large stretches within protein sequences are not involved in known functional protein-protein interactions (PPIs) [4–7]. In fact, previous studies have shown that a significant fraction of the protein interactome does not map onto any known function. This leads to an interesting question: can regions within proteins not involved in functional PPIs influence the process of formation of functional complexes? At biologically relevant concentrations, could non-interfacial residues have co-evolved to tune the binding affinities and kinetics of protein

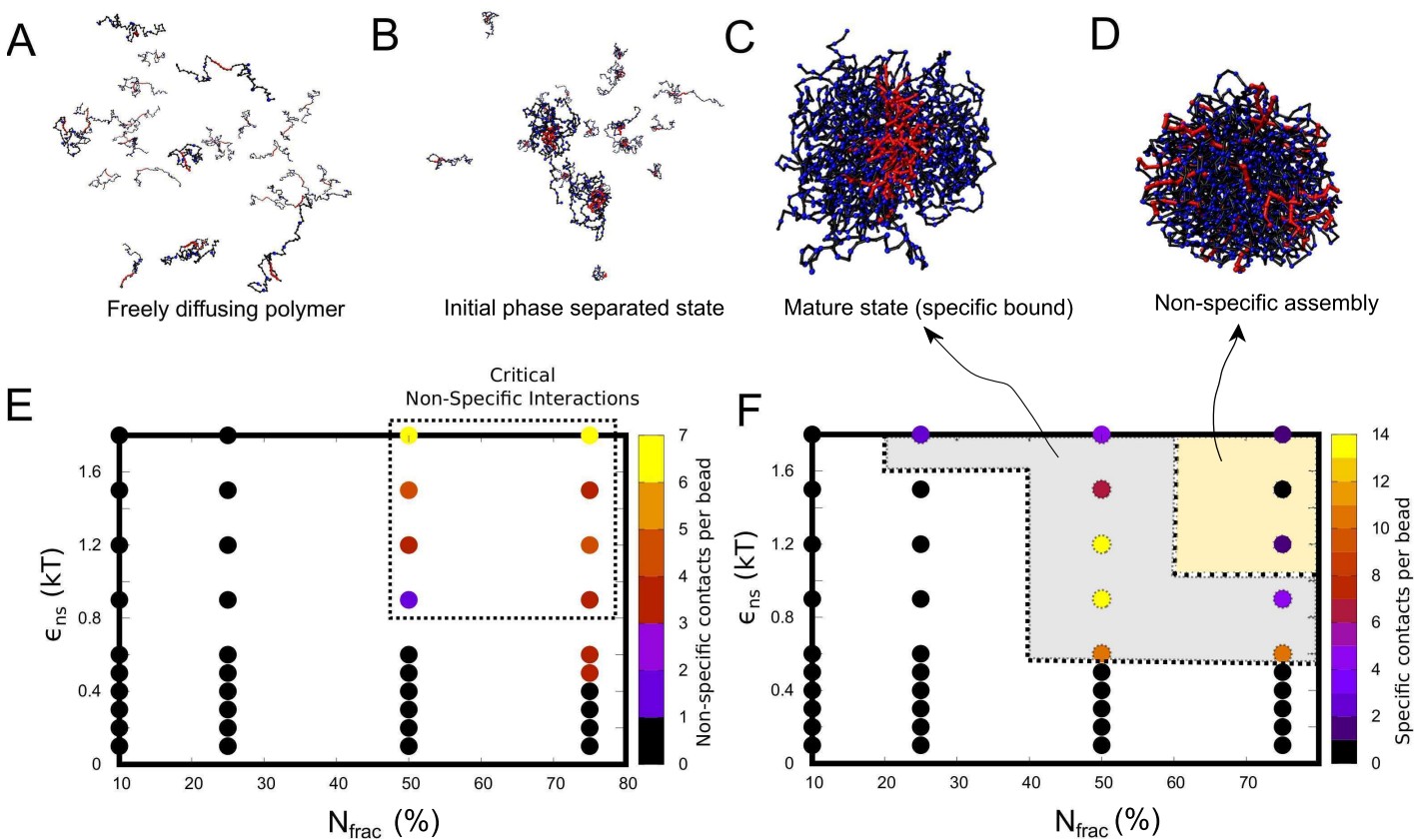

**Fig 8. Varying fraction of non-specifically interacting residues.** Snapshots of A) freely diffusing polymer in the simulation. B) early non-specifically bound initial multimer state, C) solid like mature state and D) non-specifically assembled state for a 3-bead polymeric system. E) Non-specific contacts, $N_{ns}$, per bead is plotted for varying non-specific interaction $\epsilon_{ns}$ and fraction of non-specifically interacting beads ($N_{frac}$, given in percentage). The dotted box indicate the critical values of non-specific interaction and $N_{frac}$, fraction of non-specific beads required for formation mature multimer complex. F) Specific contacts, $N_{sp}$, as a function of varying non-specific interaction, $\epsilon_{ns}$, and fraction of non-specifically interacting beads. The grey area covered inside dotted line is the region where the formation of multimer leading to mature complex is plausible. The simulations were performed for a fixed specific interaction strength, $\epsilon_{sp}$ of 1.2 $k_BT$.

complex formation? How does the interplay between specific and promiscuous interactions shape the fate of these complexes? Given the sheer size of the proteome and the vast number of protein-protein interactions that underlie cellular processes, phenomenological models could become important tools to study the physical mechanisms underlying the process of formation of functional complexes [30, 33, 34]. Previous theoretical studies have attempted to understand the effect of promiscuous interactions on the ability of proteins to bind specifically to their partners [19]. However, there is a void in bottom-up studies elucidating the potential role of non-specific interactions in promoting the formation of mature protein-protein complexes.

### Key findings in context of biomolecular function

"Noisy", non-functional interactions make up a large fraction of the proteome [13]. Also, weak interactions involving intrinsically disordered domains facilitate LLPS of proteins resulting in an accelerated rate of reaction within these biomolecule-rich compartments [35, 36]. There-fore, we model the phenomenon of the formation of functional complexes as that of associative semi-flexible polymers, which comprise of a small functional interaction interface (10% of the residues), see Fig 1. In this scenario, when the non-specifically interacting residues outnumber the functional interaction sites significantly, the formation of functional contacts incurs a

significant entropic cost. In other words, the potential configurations which have functional interactions in place are a very small fraction of overall possible configurations. The formation of functional complexes, therefore, is a non-trivial process. Using conventional Langevin dynamics and replica exchange simulations, we probe the role of non-specific interactions in modulating the process of finding the correct, functional configurations between interacting biomolecules.

Using phenomenological simulations, we identify a narrow range of non-specific interaction strength that favors the formation of functional complexes (dimers, multimers with specific interactions satisfied). Using simulations modeling dimeric complex formation, we show that there exists an optimal regime of non-specific interaction strength which allows the interaction partners to dwell long enough to find the correct functional configuration (with specific contacts formed). Even in the scenario where the specifically interacting residues are clustered together, forming a stable interaction core, the polymers have to dwell long enough in contact to 'find' the functional configuration where these patches are in contact. (Fig 2). If the non-specific interaction strengths are weaker than this optimal range ($\epsilon_{ns} \approx 0.5 \, k_B T$, also see homopolymer phase separation results in S2 Fig), the process of finding the functional configuration becomes orders of magnitude slower (Fig 3A and 3B). On the other hand, as the non-specific interactions become stronger than the optimal range, the polymer chains dwell longer in the non-functional configurations, also slowing down the search process (Fig 4).

Further, in simulations where only a fraction of non-functional residues are involved in promiscuous interactions, very few non-specific interaction sites can drive the formation of specific contacts (Fig 8). In such a regime with a few interaction sites and a narrow parameter range for self-assembly, subtle modifications such as phosphorylation of these non-core residues could significantly alter the phase behavior, critical concentrations, and dynamicity of the self-assembled structures [18, 37–39]. Conversely, few mutations resulting in extremely 'sticky' non-specific regions might lose the polymers' ability to form functional complexes. These findings imply that non-functional regions of the proteome could, therefore, have evolved under a strong, energetic constraint that requires the functional complexes to be thermodynamically favorable and assumes functionally-competent configurations at biologically relevant timescales.

## Deviation from conventional spacer-sticker models

Our modeling approach is similar in philosophy to spacer-sticker models used to study self-assembly by multi-valent biomolecules [15, 40]. Previous sticker-spacer models such as the one by Harmon et al. assume that adhesive interactions exist between sticker regions (specific interactions) alone and thereby estimate critical sticker-sticker interactions to be around 3–5 $k_B T$ [16]. In our work, we show that in the presence of an optimal, weak non-specific interaction, critical specific interaction strengths could be as low as 1.2–1.5 $k_B T$ (pairwise). Further, in our simulations, we find that for a specific interaction strength of around 1.2 $k_B T$, this optimal value of non-specific interaction is around 0.5 $k_B T$. Also, this optimal range of non-specific interactions could result in favorable kinetics of complex formation, suggesting that spacer regions could evolve independently, tuning the kinetics of complex formation (Fig 3). Our results support the recent findings by Holehouse et al., which demonstrate spacer-spacer interactions to be key driving forces of phase-separation [25].

Another key deviation from the typical spacer-sticker formalism is in how sticker-sticker interactions are modeled in our study. Unlike a fixed valency (typically 1) sticker-sticker interaction, we model sticker interactions (specific contacts in our model) as isotropic interactions with a valency of the order of the coordination number for the beads (defined by steric

constraints). While a deviation from the traditional spacer-sticker approach, this assumption is reasonable because the key objective of the current work was to understand the role of non-specific (interactions involving spacers) interactions in driving functional complex formation. Crucially, even in this limit of high specific interaction valency, an optimal strength of non-specific interactions emerges that allows specifically interacting regions of proteins to find themselves and form stable contacts at biologically relevant timescales. In the limit of low specific interaction valency (of 1 or 2), the maturation timescales (time taken for specific interactions to form) for dimers (and multimers) would show an even stronger dependence on the strength of non-specific interactions. Since non-specific interactions allow the polymers to dwell in contact long enough for the specifically interacting regions to find themselves and lock in place, a finite interaction valency would make their role even more prominent. This dependence of specific interaction maturation times on non-specific interaction strengths has previously been highlighted in a previous simulation study on sticker-spacer proteins by Ranganathan and Shakhnovich [40]. However, the choice of isotropic specific interactions results in a 'core-shell' sub-cluster organization, a feature that is an outcome of the model assumption. While the role of non-specific interactions in promoting specific contacts would largely hold true for finite valency specific interactions, the architecture of the assembly would depend on valency.

## Conclusion

Weak, promiscuous interactions between non-functional regions of biopolymers can result in loosely held, dynamic self-assemblies within which functional interfaces can find each other at biologically relevant timescales. The current study provides a mechanistic basis for the narrow range of non-specific interaction strengths that promote protein-protein interactions in proteins with a high degree of disorder (see Fig 9). The role of non-specific interactions in promoting the maturation of protein complexes with defined structural folds and lower conformational flexibility is to be further probed.

## Methods

Despite the growing body of experimental literature on biomolecular phase separation, the multi-component nature of the problem makes it hard to unravel experimentally. Hence, Molecular dynamics simulations and computational models are very valuable and have been traditionally used to understand various aspects of biomolecular structural dynamics [41]. However, the scale of this problem makes it hard to probe systematically using conventional atomistic simulations. The widespread instances of the phenomenon mean that it can be treated as a problem of self-assembling polymer chains. To understand the contribution of noisy interactions in driving the formation of mature polymer complexes, we developed a semi-flexible polymer model [42–46]. In order to study the process of macromolecular chains finding each other, we performed Langevin dynamics simulations. In our study, we consider two flexible polymer chains in a dimer simulation and 30 chains in a multimer simulation; each chain is made up of 7 red beads stretch in the center and other blue/black beads in a cubic box with periodic boundary conditions. The red beads represent residues involved in native functional interactions, while blue beads represent residues with the non-functional interaction Fig 1.

Polymer chains of length 80 beads are bonded via the harmonic potential with energy,

$$E_{stretching} = k_s \sum_{i=1}^{M-1} (|\tilde{r}_i - \tilde{r}_{i+1}| - r_0)^2, \tag{1}$$

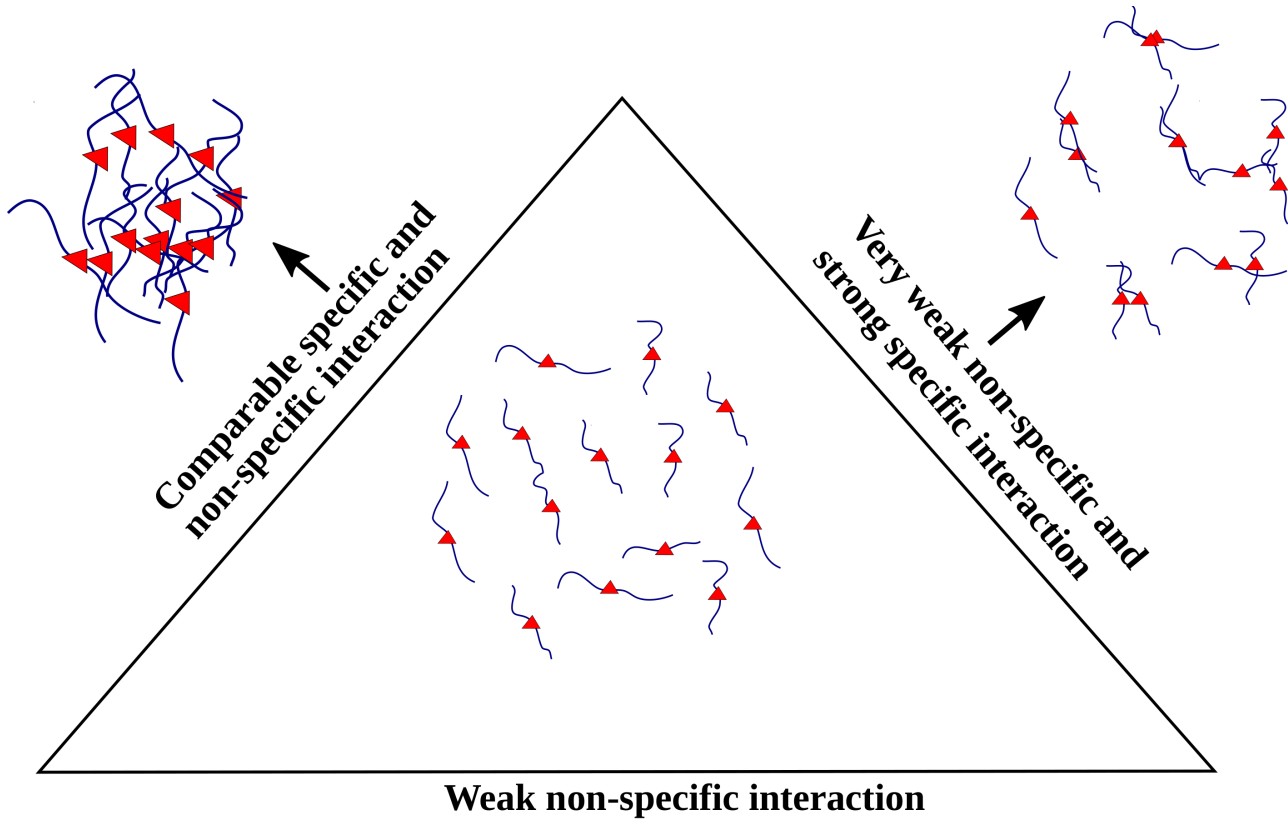

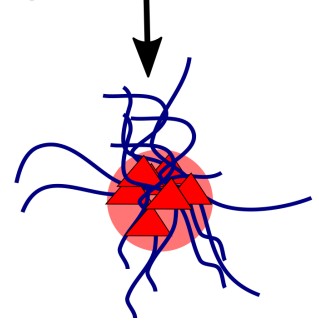

**Fig 9. Schematic representation of the overall summary of the simulations.**

where $\tilde{r}_i$ and $\vec{r}_{i+1}$ refer to $i^{th}$ and $(i+1)^{th}$ bead positions, respectively; $r_0$ refers to the equilibrium bond length and $k_s$ represents the spring constant and the value taken for $k_s$ in the simulation was 10 kcal/mol. This interaction ensures the connectivity between the beads of a polymer chain. To model bending rigidity, any two neighboring bonds in a polymer interact via the following potential

$$\mathrm{E}_{\mathrm{bending}} = \kappa \sum_{i=1}^{M-2} (1 - \cos\theta_i), \tag{2}$$

where $\theta_i$ refers to the angle between $i^{th}$ and $(i+1)^{th}$ bond, and $\kappa$ is the bending stiffness and $\kappa$ = 2 kcal/mol. All other non-bonded, inter-bead interactions were modeled using the Lennard-

Jones (LJ) potential,

$$E_{nb} = 4\epsilon \sum_{i<j} \left[ \left( \frac{\sigma}{|\tilde{r}_i - \tilde{r}_j|} \right)^{12} - \left( \frac{\sigma}{|\tilde{r}_i - \tilde{r}_j|} \right)^{6} \right],$$
(3)

for all $|\tilde{r}_i - \tilde{r}_j| < r_c$, where $r_c$ refers to the cutoff distance beyond which the non-bonded potentials are neglected. The cutoff for the LJ calculations was kept at 2.5 times $\sigma$, where $\sigma = 4.5$ Angstroms. Note that this function has two parts: a repulsive part that models steric repulsion when the beads overlap and an attractive part otherwise. For ease of implementation, in this model, this attractive part is used as an *effective interaction* that accounts for all attractive short-range forces. $\epsilon$ signifies the strength of the attractive interaction and has the units of energy. In our study, we choose different values of $\epsilon$ for interactions between red-red beads, as compared to those involving the blue beads. Since the interactions involving the blue beads model weak, non-specific contacts, they are lower in magnitude than the red-red interactions, which describe mature, native contacts. We systematically vary these two interactions to understand their role in driving the formation of mature complexes, keeping $\epsilon_{sp} > \epsilon_{ns}$. Note that the bending energy parameter ($\kappa$) controls the stiffness of the individual polymer chains, while the LJ potential parameter ($\epsilon$) controls the interaction between any two beads (inter-chain and intra-chain).

## Simulation parameters

In our study, we used the LAMMPS molecular dynamics package to perform the dynamic simulations [47], where the simulator solves Newton's equations with viscous force and a Langevin thermostat, ensuring an NVT ensemble with temperature T = 310 K. An integration timestep (dt) of 30 fs was used for the simulations. Polymer chains of length M = 80 beads, and the mass of each bead were considered to be 110 Da (average mass of amino acids). The size parameter for the beads is taken as $\sigma = 4.5$ Å. The parameters for the bonded springs was fixed as $r_0 = 4.5$ Å and $k_s = 10$ kcal/mol. The damping time for the Langevin thermostat was 1.2 ps. Similar values for these parameters have been previously used to perform coarse-grained protein simulations by Bellesia et al. and Bieler et al. ([48, 49]). All the snapshots shown in the manuscript were taken using Visual Molecular Dynamics (VMD) software [50]. The simulation scripts, data files, and analysis scripts are given in the GitHub link: https://github.com/rakeshkrish/LAMMPS-files.

## Replica exchange Langevin dynamics

In order to calculate the free energy profiles for the dimeric and multi-polymer systems (Figs 2C and 6), we employed the parallel tempering Langevin dynamics technique. In this approach, 24 independent, non-interacting copies (replicas), each at different temperatures between 309 K—401 K each 4K apart, were simulated under the NVT ensemble. In order to ensure proper sampling of equilibrium states, an exchange of configurations between different replicas is attempted every 0.5 ns. In this algorithm, a successful swap of configurations between replicas (at different temperatures) occurs based on the Boltzmann probability [51, 52]. Each of the individual replicas was simulated for a timescale of 9 $\mu s$. Using the combination of conventional Molecular dynamics and the Monte Carlo algorithm, this technique has been commonly used to enable efficient sampling of the conformational space for problems such as protein folding and self-assembly.

## Supporting information

**S1 Fig. Shows results of simulations performed with two smaller patches of specific interaction sites at the tail of the polymer.**
(EPS)

**S2 Fig. Demonstrates results from homopolymer models, showing the minimal strength of interaction for self-assembly.**
(EPS)

**S3 Fig. Shows the spatial density profile plotted in linear scale.**
(EPS)

**S1 Movie. Shows polymer chains self-assembling to the mature state.**
(MP4)

**S2 Movie. Shows self-assembly into a non-specifically assembled cluster.**
(MP4)

## Acknowledgments

The authors would like to thank Prof. B.J. Rao, Prof. Dibyendu Das, and Prof. M.Muthukumar for their suggestions and discussions related to the work. We also thank Sangram Kadam, Guhan K.A., Vinoth M., and Shuvadip Dutta for their input while writing the manuscript.

## Author Contributions

**Conceptualization:** Rakesh Krishnan, Samir K. Maji, Ranjith Padinhateeri.

**Data curation:** Rakesh Krishnan, Srivastav Ranganathan.

**Formal analysis:** Rakesh Krishnan, Srivastav Ranganathan, Samir K. Maji, Ranjith Padinhateeri.

**Funding acquisition:** Ranjith Padinhateeri.

**Investigation:** Rakesh Krishnan, Srivastav Ranganathan, Ranjith Padinhateeri.

**Methodology:** Rakesh Krishnan, Srivastav Ranganathan, Ranjith Padinhateeri.

**Project administration:** Ranjith Padinhateeri.

**Resources:** Ranjith Padinhateeri.

**Software:** Rakesh Krishnan, Srivastav Ranganathan.

**Supervision:** Ranjith Padinhateeri.

**Validation:** Srivastav Ranganathan, Ranjith Padinhateeri.

**Visualization:** Rakesh Krishnan, Srivastav Ranganathan, Ranjith Padinhateeri.

**Writing – original draft:** Rakesh Krishnan, Srivastav Ranganathan, Samir K. Maji, Ranjith Padinhateeri.

**Writing – review & editing:** Rakesh Krishnan, Srivastav Ranganathan, Samir K. Maji, Ranjith Padinhateeri.

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
