## [Decision Letter · Decision Letter 0]

10 Nov 2021

Dear Dr Ranganathan,

Thank you very much for submitting your manuscript "Role of non-specific interactions in the phase-separation and maturation of macromolecules" for consideration at PLOS Computational Biology.

As with all papers reviewed by the journal, your manuscript was reviewed by members of the editorial board and by several independent reviewers. In light of the reviews (below this email), we would like to invite the resubmission of a significantly-revised version that takes into account the reviewers' comments.

We cannot make any decision about publication until we have seen the revised manuscript and your response to the reviewers' comments. Your revised manuscript is also likely to be sent to reviewers for further evaluation.

Sincerely,

Avner Schlessinger

Associate Editor

PLOS Computational Biology

Arne Elofsson

Deputy Editor

PLOS Computational Biology

Reviewer's Responses to Questions

**Comments to the Authors:**

Reviewer #1: The manuscript discusses the phase separation of linear polymers, made of a non-specifically-interacting chain and featuring a few specifically interacting sites located at the center of the chain. The attention is put on whether polymers are able to phase separate and particularly to form a "mature" condensate, where contacts among specifically interacting sites are maximized. The main finding is that weak non-specific interactions facilitate phase-separation and maturation, while strong non-specific interactions favor a non-functional gel state. Tweaking attraction (from absent, to weak, to strong) throughout the polymer chain could allow fine tuning of phase-separation properties in biological contexts.

I think the paper is well structured and presents sound and valuable results, that support the conclusions (with the few minor exceptions below). The manuscript is of certain interest to the community of physicists, chemists and biologists working on biological phase-separation phenomena. In my opinion, its main defects are a not well circumstantiated description of the methods, essential for reproducibility, and, most importantly, an incomplete discussion of the effective physiological relevance of the model. Readability can also be improved. Overall, I would be happy to recommend publication if the authors manage to resolve my concerns, detailed below in decreasing order of importance.

1) In the model, red-red interactions seem to be non-exclusive (i.e. one red bead, or region of red beads, can interact with more than one other red bead, or region of red beads). This is an important limitation when modeling specific interactions, that, to my knowledge, are selective and mostly exclusive (e.g. nucleic acids or SH3 and proline-rich domains). This distinguishes them from non-specific interactions, that are less selective and not exclusive (e.g. hydrophobic disordered domains). Many other computational works take this into account when modelling specificity. It might be a matter of semantics, but the authors should clarify this and discuss the relevance of their model, providing, if possible, biological examples.

Another feature of the model that calls for a justification is the fact that all specifically-interacting sites are placed at the center of the polymer and they are always grouped together. Without a comment on how this assumption influences the results, this might look artificial and limit relevance.

2) I wonder whether the gel-like state obtained at large non-specific interaction can really be called "kinetically trapped". To me, a kinetically trapped state is one that corresponds to a local minimum of free energy, separated from the absolute minimum by a prohibitive barrier. According to the measurements of free energy presented in the manuscript, in this regime the system does seem to be at a free energy absolute minimum (see e.g. Fig. 3C, orange curve). This suggests that increasing epsilon_ns changes the free energy landscape and simply makes the gel phase more favorable, which seems to me a more correct statement. If so, it would be interesting to comment on why this happens: maybe the configurational entropy of the large section of blue beads overcomes the slight energetic advantage that a state with clustered red beads might bring?

3) The methods section is missing details on how T_dwell, T_mature and R_g^norm are computed. Additionally, it does not provide the value of kappa (bending rigidity parameter) used in simulations and, most importantly, the criterion used for its choice. Table 2 is missing the strength of blue-red interaction.

4) I think the first part of the introduction should be toned down a bit. In particular:

- "[...] the local concentration of proteins within these compartments is 50-100 fold higher than their bulk concentrations [3]."

This statement is not as general as it seems. There are plenty of counter-examples where condensates have a partitioning fraction of less than 50. See for instance the introduction of

Ghosh, A., Mazarakos, K. & Zhou, H. Three archetypical classes of macromolecular regulators of protein liquid–liquid phase separation. Proc. Natl. Acad. Sci. 116, 19474–19483 (2019),

but also, for specific examples

Sanders, D. W. et al. Competing Protein-RNA Interaction Networks Control Multiphase Intracellular Organization. Cell 181, 306-324.e28 (2020).

Banani, S. F. et al. Compositional Control of Phase-Separated Cellular Bodies. Cell 166, 651–663 (2016).

Fritsch, A. W. et al. Local thermodynamics govern formation and dissolution of Caenorhabditis elegans P granule condensates. Proc. Natl. Acad. Sci. 118, e2102772118 (2021).

- "could liquid-like phase-separated compartments facilitate biochemical reactions within this ‘polymer-rich’ phase by increasing the likelihood of key functional interactions?"

This point is presented as a new open question, but actually derives from the commonly accepted belief that phase-separated compartments are key to accelerate physico-chemical reactions (see Ref. 32, for instance).

- The authors claim that their results "suggest that non-specific interactions are essential in the formation of phase-separated states stabilized by specific interactions."

Such a strong statement might need some clarification and comparison with the literature. Restricting ourselves to the literature of computational coarse-grained models, there are plenty of examples where phase-separating systems are successfully obtained by modeling only specific interactions (or bonds), both in patchy-particle models and in polymer-based models.

5) I wonder whether the curves in Fig. 5A (panel label missing) have been rescaled to correct for spherical geometry, so as to show a proper density. This would seem to me the best way to present the data. This also possibly solves my issue about why we see peaks (especially in the orange curve), rather than a constant uniform density that rapidly goes to 0 for distances larger than the size of the droplet.

Also, the authors should clarify how the red curve is computed, given that clusters do not form for such small interaction strength.

6) A few suggestions for readability:

- Fig. 1A and B: I suggest adding double arrows ( < ) between particles in the legend at the bottom. Also "no interaction" was a bit misleading to me, because volume exclusion is still present; steric hindrance is mentioned in the text, but it might be worth clarifying the figure as well, or at least the caption.

- Page 3 mentions a "model section", but there is no such section.

- At the beginning of page 4, I suggest making clear in the text that those simulations are run with only 2 polymers.

- The notation of <n_np> and <n_ns> is misleading: at places, symbols of thermodynamic averages are dropped, which is puzzling.

- Fig. 4: It would be nice to see panels C and D starting from time=0. This might possibly solve my question about why N_ns starts from 12 in C and from the much higher value of 20 in D.

- Fig. 8E and F: I suggest including the % symbol in the horizontal axis label.

- Methods: it's unclear why red and blue beads are defined as 'X' and 'Y', tags that are never used in the manuscript except for that one sentence.

- Fig. 9 is not cited.</n_ns></n_np>

Reviewer #2: Krishnan et al. study the role of non-specific interactions in the formation of biomolecular condensates. The authors present convincing results concerning the important role that non-specific interactions play in condensate formation. These are important and timely results. However, some of the model choices should be better justified. In addition, the transition in material state reported by the authors, from solid-core to glass, is not clear. We usually see a transition from a liquid to a solid, not from an aggregate (for lack of a better term) to a glass state. Finally, the strength of non-specific interactions should be discussed with respect to what occurs in biological systems.

Below are specific points that should be addressed:

1. The following model decisions should be explained:

a. Why 80-beads long and the choice to represent each protein as a worm-like chain (semi-flexible polymer),

b. Each chain contains red beads participating in specific interactions and blue beads participating in non-specific interactions. How is the ratio determined? Is that based on your statement from page 4: “interface residues involved in stabilizing biomolecules complexes account for only a tiny fraction of protein surfaces” (which is missing references)?

c. Why were the red beads placed in the center of the polymer, and would changing that affect the conclusions?

d. Please confirm that blue beads can non-specifically bind red beads.

2. “Dimers where the central red patches are in contact are referred to as mature dimers” / “dimers that are stabilized by non-specific contacts … kinetically trapped dimers” (page 4). Are those determined in steady-state?

3. In FIg. 2, the division of the panel into three regimes (monomer, mature dimer, and kinetically trapped dimer) should be clarified. Does that mean that points that are within the purple region are kinetically trapped? If that is the case, how come the ratio of e_ns to e_sp does not change the type of dimer, only the number of contacts. Also, please clarify whether the endpoint of these simulations is at steady state.

4. Please explain in Methods how Tdwell and Tmature were calculated.

5. What is the realistic ratio between specific and non-specific interactions? In Fig. 3, for example, in some of the conditions, the energy of the specific interactions is the same as the non-specific ones. Please provide some discussion on the ratio expected in biological systems between specific and non-specific interactions and whether you see the non-specific (red/blue) computing out the specific ones (red/red).

6. “These results suggest that there exists an optimal range of ens which enables the dimers to remain stable long enough so as to allow maturation to occur” - is this a range we see in biological systems? Please discuss.

7. In the condensate simulations discussed on page 6, I am not sure why mature condensate translates to having all the read beads clustered in the center. This is actually different from what we observe in condensates expressed either in vitro or in cells. What is the valency of each patch, and how come steric clashes do not play a role here? Movies of these simulations would help in understanding these dynamics.

8. “It is interesting to note that the non-specific interaction leads to a phase separation-like phenomena further transitioned to a mature state.” (page 8). Please explain how you define “phase separation-like.” Also - do you mean that the solid-like core transition into a glass-like state? This is not clear and needs further explanation.

9. With regards to Fig. 5 and the text describing it: it is not clear how a stable core of dimers suddenly rearranges itself into a glass-like state. Is that only with temperature change or other system perturbation?

10. Fig. 7B: not clear where are the Nsp plots.

11. Fig. 7 shows an important point - that non-specific binding demixes type 1 from type 2 proteins. What is the minimal e_ns needed to achieve this behavior? In system 2 - do you see type 1 join the condensate before type 3 or no preference in assembly order?

12. Fig. 8: what is the minimum percentage of specific interactions needed to form a condensate? The authors show 20% as the minimal percentage, but can that go even lower? What happens at 100% non-specific interactions?

13. Please make the simulations results and code available to the reader via github or similar platform.

Few typos:

1. “These results suggests” (page 5)

2. “As discussed in the model section” (page 3), there is no model section

3. “As described so far (see Fig. 1B)”, I assume the authors mean Fig. 1A

4. Table 2, default row: you are missing the energy term for blue-red interactions.

5. No reference to Figure 7A

**Have the authors made all data and (if applicable) computational code underlying the findings in their manuscript fully available?**

Reviewer #1: **No: **Data not publicly available (yet?)

Reviewer #2: **No: **Simulation code and results should be publicly available.

PLOS authors have the option to publish the peer review history of their article (what does this mean?). If published, this will include your full peer review and any attached files.

Reviewer #1: No

Reviewer #2: No
---

## [Decision Letter · Decision Letter 1]

7 Mar 2022

Dear Dr Ranganathan,

Thank you very much for submitting your manuscript "Role of non-specific interactions in the phase-separation and maturation of macromolecules" for consideration at PLOS Computational Biology. As with all papers reviewed by the journal, your manuscript was reviewed by members of the editorial board and by several independent reviewers. The reviewers appreciated the attention to an important topic. Based on the reviews, we are likely to accept this manuscript for publication, providing that you modify the manuscript according to the review recommendations.

Sincerely,

Avner Schlessinger

Associate Editor

PLOS Computational Biology

Arne Elofsson

Deputy Editor

PLOS Computational Biology

[LINK]

Reviewer's Responses to Questions

**Comments to the Authors:**

Reviewer #1: The authors answered most of my queries. The paper has improved and now the model is much clearer and its implications better interpreted. The relevance of the assumptions made is still open for discussion, especially for what concerns exclusivity, in the sense of finite valence of specifically interacting sites. But the reader is now given better tools to assess significance on their own. In addition, the new Supplementary Figures help conveying the idea that results could qualitatively hold also for different polymer structures.

I am still puzzled by Fig. 5A: if R1 and R2 are varied in such a way to keep their distance constant to 5 angstroms, and if the cluster center of mass is at 0, what the authors describe in the Model section is really the definition of volume density. The computed quantity should be constant at distances smaller than the radius of the red cluster and go to 0 at larger distances. To me, the current Fig. 5A suggests that red clusters are empty shells. Indicative of high local density of red beads would be the difference between the small- and the large-distance plateau, rather than the presence of peaks. It looks as if some geometrical factor was forgotten during the binning procedure. Also Fig. 5B is wrongly labelled and, if I am right in assuming that the horizontal axis is actually epsilon_ns, the peak densities do not correspond to the ones in Fig. 5A. But again, I do not see the meaning of these peak values. Can the authors clarify what is happening here?

I also suggest to avoid interpolating data with non-linear polynomial curves, as this is misleading (Figs. 3A-B and 5B).

Finally, the abstract still refers to "kinetic trapping" and on page 4 "inter-chain specific and non-specific contacts" should be inverted.

Reviewer #2: The authors addressed all of my concerns.

**Have the authors made all data and (if applicable) computational code underlying the findings in their manuscript fully available?**

Reviewer #1: Yes

Reviewer #2: Yes

PLOS authors have the option to publish the peer review history of their article (what does this mean?). If published, this will include your full peer review and any attached files.

Reviewer #1: No

Reviewer #2: No

Figure Files:

Data Requirements:

Reproducibility:

References:

---

## [Decision Letter · Decision Letter 2]

29 Mar 2022

Dear Dr Ranganathan,

We are pleased to inform you that your manuscript 'Role of non-specific interactions in the phase-separation and maturation of macromolecules' has been provisionally accepted for publication in PLOS Computational Biology.

Best regards,

Avner Schlessinger

Associate Editor

PLOS Computational Biology

Arne Elofsson

Deputy Editor

PLOS Computational Biology

Reviewer's Responses to Questions

**Comments to the Authors:**

Reviewer #1: The authors have addressed my comments and corrected the wrong figure. Figure 5B now makes physical sense.

**Have the authors made all data and (if applicable) computational code underlying the findings in their manuscript fully available?**

Reviewer #1: None

PLOS authors have the option to publish the peer review history of their article (what does this mean?). If published, this will include your full peer review and any attached files.

Reviewer #1: No

---

## [Editor Report · Acceptance letter]

1 May 2022

PCOMPBIOL-D-21-01707R2 

Role of non-specific interactions in the phase-separation and maturation of macromolecules

Dear Dr Ranganathan,

I am pleased to inform you that your manuscript has been formally accepted for publication in PLOS Computational Biology. Your manuscript is now with our production department and you will be notified of the publication date in due course.

With kind regards,

Andrea Szabo
